# Deterministic Error Bounds for Euclidean Clustering

## Abstract

This paper gives a closed-form solution to Euclidean clustering, also known as k-means clustering. The key observation behind our solution is that the features of clustered data lie near a subspace whose projection operator encodes the clustering. In contrast to classical alternating approaches like Lloyd's algorithm or K-means++, which suffer local minima, our solution can be trivially computed with a singular value decomposition. Moreover, we show that if the distinct clusters are sufficiently well-defined (meaning different clusters are sufficiently separated, and data in each cluster not too scattered), our solution is deterministically guaranteed to be correct. We corroborate our theoretical findings with a comprehensive array of experiments, showing that simple relaxations of our solution yield algorithms that not only rival but also surpass the current state-of-the-art in a wide variety of settings, both in terms of accuracy and speed.

## 1 Introduction

Euclidean clustering is a fundamental learning task, with applications spanning robotics, computer vision, environmental modeling, and many more (Ahmed et al., 2020; Ma et al., 2023). In words, this type of clustering involves identifying groups of data points that are close to each other in Euclidean space with respect to some metric, typically the $\ell_2$-norm. Predominantly, this task is performed through variants of Lloyd's algorithm (Lloyd, 1982), such as K-means++ (Arthur & Vassilvitskii, 2007), or spectral clustering (Ng et al., 2001). In essence, these algorithms operate iteratively, starting with the selection of K initial cluster centroids. Then, in each iteration, these algorithms alternate between two main steps: (i) assigning each data point to its nearest centroid, forming clusters, and then (ii) recalculating the centroids as the mean of all the points assigned to each cluster. This process is repeated until convergence, typically stopping when the centroids no longer change significantly or after a specified number of iterations. The main idea behind this approach is to try to minimize the sum of squared distances between data points and their assigned centroids. Unfortunately, the alternating nature of these methods induces local minima, meaning that they heavily depend on initialization. Due to this and other shortcomings in existing methodology, coupled with the cornerstone applicability of this learning task, Euclidean clustering remains an active and vibrant area of research (see (Liu et al., 2017; He & Zhang, 2018; Wang et al., 2019; Lee, 2019; Yang et al., 2018; Wu & Peng, 2017; Ma et al., 2023; Ahmed et al., 2020; Chen & Witten, 2023; Huang et al., 2023; Hu et al., 2023; Laber et al., 2023) and references therein for yet a very incomplete list of related work).

### Contributions

This paper derives a closed-form solution to Euclidean clustering. As opposed to the classical alternating approaches outlined above, our solution leverages the structure induced in the feature space when the data forms clusters. As it turns out, the features of clustered data lie near a subspace, and the projection operator of such subspace encodes the information about the clustering in the form of an adjacency matrix from which the clustering can be revealed by inspection after a simple threshold operation. Since projection operators are unique, the subspace in question and the clustering can be learned through a simple singular value decomposition.

In contrast to traditional local minima guarantees, our main theoretical result shows that if the clusters are well-defined, our solution is deterministically guaranteed to be correct. More precisely, we

demonstrate that if the clusters are sufficiently separated from one another, and the data in each cluster is not too scattered, then our closed-form solution will identify the clusters perfectly with probability one. Intuitively, this notion of clusters distinctness can be thought of as a proxy for noise. Under this view, our results guarantee correctness as a function of the noise level, but independent of its distribution (normal, poisson, etc.). Moreover, the proof of this result is quite elegant, and can be obtained through a few clever observations, and an application of the Davis-Kahan $\sin(\boldsymbol{\Theta})$ Theorem (Davis & Kahan, 1970; Stewart & Sun, 1990).

We point out that certifying the output of an algorithm may appear of secondary importance in some applications, such as predicting preferences or ratings in recommender systems (Chen et al., 2009; Recht, 2011). However, there are more sensitive applications where the reliability of an output is paramount. One example of such application is single-cell sequencing (Klimovskaia et al., 2020), which aims to identify correlations between active genes and traits of different types of cells (bone, muscle, brain). Lloyd's algorithm and its variants related to orthogonal nonnegative matrix factorization (Ang & Gillis, 2019; Li & Ding, 2018; Gan et al., 2021) are among the most prevalent algorithms used to reveal these correlations. However, since no practical algorithm provides correctness guarantees, researchers are often forced to turn to alternative heuristic accuracy indicators such as the silhouette index or the Davies-Bouldin index Ashari et al. (2023), which may produce conflicting results, and may lead to spurious conclusions. In other words, the results obtained from existing algorithms are suspect. This work provides addresses this shortcoming.

We complement our theoretical findings with a comprehensive array of experiments, both on synthetic data, and on five real datasets, precisely related to single-cell sequencing. Besides verifying our analysis, these results also show that simple relaxations of our closed-form solution yield practical algorithms that degrade nicely with noise. In fact, these variants not only rival but also surpass the current state-of-the-art in a wide variety of settings, both in terms of accuracy and speed.

ORGANIZATION OF THE PAPER

The rest of the paper is organized as follows. Section 2 gives a description of the problem setup, and a brief overview of related work, which will give context to our main contributions in Section 3. This is followed by the exposition of our key ideas and the proof of our theoretical findings in Section 4. Later, Section 5 discusses adaptations that can be made to our closed-form solution to obtain relaxations with practical advantages. Finally, Section 6 presents our experiments.

## 2  PREAMBLE AND RELATED WORK

Let $\{\mathbf{x}_i\}$ be a collection of $n$ samples in $\mathbb{R}^m$. Let $\{\Omega_k^\star\}$ be a partition of $\{1,\ldots,n\}$ indicating the *true* clustering of the samples among $K \leq \min(m,n)$ groups. Suppose

$$\mathbf{x}_i \;=\; \sum_{i=1}^{n} \boldsymbol{\mu}_k^\star \mathbb{1}_{\{i \in \Omega_k^\star\}} + \mathbf{z}_i, \tag{1}$$

where $\boldsymbol{\mu}_k^\star \in \mathbb{R}^m$ is the *true centroid* of the $k^{\text{th}}$ cluster, $\mathbb{1}$ denotes the indicator function, and $\mathbf{z}_i \in \mathbb{R}^m$ determines the separation of $\mathbf{x}_i$ from its corresponding centroid, which can be considered as noise. Given $\{\mathbf{x}_i\}$, our goal is to estimate $\{\Omega_k^\star\}$ and $\{\boldsymbol{\mu}_k^\star\}$. This is typically pursued through the following optimization

$$\min_{\{\Omega_k, \boldsymbol{\mu}_k\}} \sum_{k=1}^{K} \sum_{i \in \Omega_k} \|\mathbf{x}_i - \boldsymbol{\mu}_k\|, \tag{2}$$

where $\|\cdot\|$ is any suitable metric, usually the $\ell_2$ norm, in which case (2) minimizes the intra-cluster variance. Traditionally, (2) is sought through variants of Lloyd's Algorithm (Lloyd, 1982), which initializes centroid estimates $\{\hat{\boldsymbol{\mu}}_k\}$, and alternates between solving the following problems until convergence

$$\hat{k}_i = \arg\min_k \|\mathbf{x}_i - \hat{\boldsymbol{\mu}}_k\|, \qquad\qquad \hat{\boldsymbol{\mu}}_k = \arg\min_{\boldsymbol{\mu}_k} \sum_{\substack{\mathbf{x}_i: \\ \hat{k}_i = k}} \|\mathbf{x}_i - \boldsymbol{\mu}_k\|.$$

In words, these two problems iteratively assign the $i^{\text{th}}$ sample to its closest centroid estimate, indexed by $\hat{k}_i$, and recomputes the centroid $\hat{\boldsymbol{\mu}}_k$ as the distance minimizer among the samples assigned to the $k^{\text{th}}$ group. There are numerous variants of this algorithm, including a weighted version (Liu et al., 2017), a kernel Nystrom approximation (He & Zhang, 2018), a fast adaptive version (Wang et al., 2019), a non-alternating stochastic version (Lee, 2019), deep learning methods (Yang et al., 2018; Wu & Peng, 2017), and many more (Ma et al., 2023; Ahmed et al., 2020; Chen & Witten, 2023; Huang et al., 2023; Hu et al., 2023; Laber et al., 2023).

## 3 MAIN RESULTS

Here we propose a new method to learn $\{\Omega_k^\star\}$ and $\{\boldsymbol{\mu}_k^\star\}$ directly, without the need for an alternating algorithm. The key insight behind our solution is that the features of the data lie near a subspace whose projection operator $\mathbf{P}^\star$ encodes the clustering. We simply estimate such operator using a singular value decomposition, and threshold it to reveal the clustering. To see this, let $\mathbf{X} \in \mathbb{R}^{m \times n}$ be the matrix containing the columns in $\{\mathbf{x}_i\}$. Let $\mathbf{V} \in \mathbb{R}^{n \times K}$ be the matrix containing the $K$ leading right singular vectors of $\mathbf{X}$. Let $\hat{\mathbf{P}} := \mathbf{V}\mathbf{V}^\mathsf{T} \in \mathbb{R}^{n \times n}$ be the projection operator onto $\text{span}(\mathbf{V})$. Define $\hat{\mathbf{P}}_\lambda$ as the entry-wise thresholded version of $\hat{\mathbf{P}}$ with entries

$$[\hat{\mathbf{P}}_\lambda]_{ij} := \left\{ \begin{array}{ll} \hat{\mathbf{P}}_{ij} & \text{if } |\hat{\mathbf{P}}_{ij}| > \lambda \\ 0 & \text{otherwise,} \end{array} \right.$$

where $\lambda \in [0, 1]$ is a parameter that depends on the positions of $\{\boldsymbol{\mu}_k^\star\}$ and the intra-cluster dispersion level determined by $\{\mathbf{z}_i\}$. Our estimates are the distinct supports (i.e., non-zero patterns) of the columns in $\hat{\mathbf{P}}_\lambda$, denoted as $\{\hat{\Omega}_k\}$, and

$$\hat{\boldsymbol{\mu}}_k := \arg\min_{\boldsymbol{\mu}_k} \sum_{i \in \hat{\Omega}_k} \|\mathbf{x}_i - \boldsymbol{\mu}_k\|. \tag{3}$$

For example, in the special case of the $\ell_2$ norm, $\hat{\boldsymbol{\mu}}_k$ simplifies to $\frac{1}{n}\sum_{i \in \hat{\Omega}_k} \mathbf{x}_i$.

Our main theoretical result shows that these estimates are deterministically correct as long as the samples in each cluster are not too scattered relative to the location of the centroids. In order to present this result, let $\boldsymbol{\mathcal{M}}^\star \in \mathbb{R}^{m \times K}$ be the matrix containing the columns in $\{\boldsymbol{\mu}_k^\star\}$. Let $\boldsymbol{\Omega}_k^\star \in \{0, 1\}^n$ be the vector whose $i^{\text{th}}$ entry is equal to 1 if $i \in \Omega_k^\star$, and equal to zero otherwise. In words, $\boldsymbol{\Omega}_k^\star$ is the binary vector indicating the samples that belong to the $k^{\text{th}}$ cluster. Let $\boldsymbol{\Omega}^\star \in \{0, 1\}^{n \times K}$ and $\mathbf{Z} \in \mathbb{R}^{m \times n}$ be the matrices containing the columns in $\{\boldsymbol{\Omega}_k^\star\}$ and $\{\mathbf{z}_i\}$. With this we are ready to present our main result, which we summarize in the following theorem.

---

**Theorem 1.** *Let $\delta > 0$ be the gap between the $K^{\text{th}}$ singular value of $\boldsymbol{\mathcal{M}}^\star \boldsymbol{\Omega}^{\star\mathsf{T}}$ and the $(K+1)^{\text{th}}$ singular value of $\mathbf{X}$. Suppose*

$$\delta > \sqrt{2^3 K}\|\mathbf{Z}\| \max_k |\Omega_k^\star|. \tag{4}$$

*Then the clusters $\{\hat{\Omega}_k\}$ obtained with $\lambda = \frac{1}{2 \max_k |\Omega_k^\star|}$ are identical to the* true *clusters $\{\Omega_k^\star\}$ (up to a label permutation).*

---

Intuitively, the $K^{\text{th}}$ and $(K+1)^{\text{th}}$ singular values of $\boldsymbol{\mathcal{M}}^\star \boldsymbol{\Omega}^{\star\mathsf{T}}$ and $\mathbf{X}$ can be respectively interpreted as the smallest separation between centroids, and the largest separation of a sample from its cluster. The condition on $\delta$ essentially requires that the former (distance between clusters) is large enough relative to the later (cluster width), so that the clusters are discernible from our estimator, and no sample is misclustered. In this sense, $\delta$ can be interpreted as the signal to noise ratio. The proof is in Section 4, and it essentially uses the Davis-Kahan $\sin(\boldsymbol{\Theta})$ Theorem (Davis & Kahan, 1970; Stewart & Sun, 1990) to show that the support of $\hat{\mathbf{P}}_\lambda$ will coincide with that of the *true* $\mathbf{P}^\star$ (projection operator onto the span of $\boldsymbol{\Omega}^\star$). For incoherent matrices $\hat{\mathbf{P}}$, the condition on $\delta$ can be relaxed by a factor of $\mathcal{O}(K^4/\sqrt{n})$ using the $\ell_\infty$ perturbation bound in (Fan et al., 2018) instead. In practice,

however, we see that both conditions appear to be overly sufficient, as the method succeeds under much weaker conditions (see Figure 1).

Notice that in general, $\max_k |\Omega_k^\star|$ is unknown, and therefore, so is $\lambda$. However, since Theorem 1 guarantees the existence of a parameter $\lambda$ that results in a perfect recovery of (the support of) $\mathbf{P}^\star$, $\lambda$ can be trivially found by searching for the threshold that produces a matrix $\mathbf{P}_\lambda$ with exactly $K$ distinct disjoint supports that cover all the columns and all the rows.

We point out that perfect recovery of $\mathbf{P}^\star$ is a sufficient condition for clustering, convenient for analysis, but by no means necessary, and overly strict in practice. This is confirmed by our experiments in Section 6, showing that perfect clustering is possible even if $\mathbf{P}^\star$ is not perfectly recovered. This is quite fortunate and unsurprising, because even if the support of $\hat{\mathbf{P}}_\lambda$ is different than that of $\mathbf{P}^\star$ for every $\lambda \in [0, 1]$, if the difference between $\mathbf{P}^\star$ and $\hat{\mathbf{P}}$ is not overwhelmingly large, $\hat{\mathbf{P}}$ may still contain enough information to reveal the clustering through a relaxed method. An example of such relaxation would be to agglomerate samples $(i, j)$ if $[\hat{\mathbf{P}}_\lambda]_{ij} > 0$ as we decrease $\lambda$ from 1 to 0. Our future work will investigate methods like this, their properties, and their guarantees.

Finally, notice that under the conditions of Theorem 1, our estimators $\{\hat{\boldsymbol{\mu}}_k\}$ are the optimal estimators of $\{\boldsymbol{\mu}_k^\star\}$. We formalize this in the following corollary

**Corollary 1.** *Under the same assumptions of Theorem 1, and up to a label permutation, our closed-form estimators $\{\hat{\boldsymbol{\mu}}_k\}$ satisfy*

$$\hat{\boldsymbol{\mu}}_k = \arg\min_{\boldsymbol{\mu}_k} \sum_{i \in \Omega_k^\star} \|\mathbf{x}_i - \boldsymbol{\mu}_k\|. \tag{5}$$

The proof of Corollary 1 follows immediately because under the assumptions of Theorem 1, $\{\hat{\Omega}_k\}$ are equal to $\{\Omega_k^\star\}$ (recall the definition of $\hat{\boldsymbol{\mu}}_k$ in (3) and compare to (5)). We point out that our solution truly minimizes the distance to the centroid (e.g., variance) of the *true* clustering. This should be contrated with the classical solution to (2) pursued by Lloyd's Algorithm and its variants, which minimizes distance with respect to *some* estimated clustering $\{\boldsymbol{\Omega}\}$, which may be correct or incorrect.

## 4 REASONING AND PROOF

The key idea behind our approach is that the rows of $\mathbf{X}$ lie near the subspace spanned by $\boldsymbol{\Omega}^\star$, whose projection operator $\mathbf{P}^\star$ encodes the clustering through its nonzero entries. Since projection operators are unique, we can estimate $\mathbf{P}^\star$. Using perturbation theory we show that under the assumptions of Theorem 1, our estimator $\hat{\mathbf{P}}$ cannot be too different from $\mathbf{P}^\star$, and so a simple thresholding of $\hat{\mathbf{P}}$ with an appropriate parameter $\lambda$ will reveal the nonzero entries of $\mathbf{P}^\star$, and in turn, the desired clustering.

To start, recall that $\boldsymbol{\mathcal{M}}^\star \in \mathbb{R}^{m \times K}$ and $\mathbf{Z} \in \mathbb{R}^{m \times n}$ contain the columns in $\{\boldsymbol{\mu}_k^\star\}$ and $\{\mathbf{z}_i\}$, and that $\boldsymbol{\Omega}^\star \in \{0, 1\}^{n \times K}$ indicates the clustering. With this, we can rewrite (1) simultaneously for every $i$ as

$$\mathbf{X} = \boldsymbol{\mathcal{M}}^\star \boldsymbol{\Omega}^{\star\mathsf{T}} + \mathbf{Z}.$$

For example, if there were $K = 3$ clusters, and the if the $k^{th}$ block of columns in $\mathbf{X}$ lie in the $k^{th}$ cluster, our construction would look like:

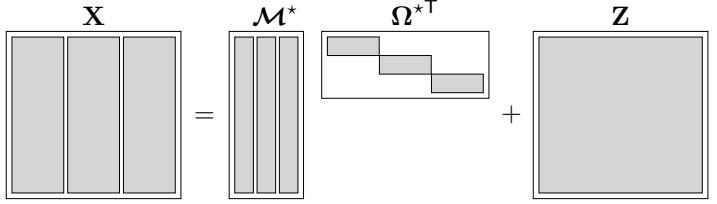

Here, the white and shaded areas represent zero and non-zero entries, respectively. It follows that the rows of $\mathbf{X}$ approximately lie in the subspace spanned by $\boldsymbol{\Omega}^\star$. We can estimate this subspace as the span of $\mathbf{V}$. Now, in general, even if $\mathbf{Z} = \mathbf{0}$, $\mathbf{V}$ may not be equal to $\boldsymbol{\Omega}$, because $\mathbf{V}$ and $\boldsymbol{\Omega}$ may be

different bases of the same subspace. However, projector operators are unique, so we can estimate $\mathbf{P}^\star$ as $\hat{\mathbf{P}} = \mathbf{V}\mathbf{V}^\mathsf{T}$ (recall that the columns of $\mathbf{V}$ contain the K leading right singular vectors of $\mathbf{X}$). Next observe that the supports of $\{\mathbf{\Omega}_k^\star\}$ are disjoint, so $\mathbf{\Omega}^\star$ is orthogonal. Hence,

$$\mathbf{P}^\star = \bar{\mathbf{\Omega}}\bar{\mathbf{\Omega}}^\mathsf{T}, \tag{6}$$

where $\bar{\mathbf{\Omega}}$ is the matrix formed with $\{\mathbf{\Omega}_k^\star/\|\mathbf{\Omega}_k^\star\|\}$ as columns. That is, $\bar{\mathbf{\Omega}}$ is equal to $\mathbf{\Omega}^\star$ except its columns are normalized. From (6) we can see that the columns and rows of $\mathbf{P}^\star$ have the exact same supports as the columns in $\mathbf{\Omega}^\star$. For example, if the first third of the samples were in one cluster, and similar for the second and third thirds, as in our construction above, the supports of a pair $(\mathbf{\Omega}^\star, \mathbf{P}^\star)$ would look like:

$$\mathbf{\Omega}^\star = \quad\quad , \quad\quad \mathbf{P}^\star = \quad\quad .$$

We will now show that the entries of our estimator $\hat{\mathbf{P}}$ cannot be too different from the entries in $\mathbf{P}^\star$. Specifically, we will show that corresponding entries in these matrices cannot differ by more than $\frac{1}{2\max_k |\Omega_k^\star|}$. To see this, write

$$\|\mathbf{P}^\star - \hat{\mathbf{P}}\|_F^2 = \|\mathbf{P}^\star\|_F^2 - 2\mathrm{tr}(\hat{\mathbf{P}}^\mathsf{T}\mathbf{P}^\star) + \|\hat{\mathbf{P}}\|_F^2 = 2K - 2\mathrm{tr}(\hat{\mathbf{P}}^\mathsf{T}\mathbf{P}^\star)$$
$$= 2K - 2\mathrm{tr}(\mathbf{V}\mathbf{V}^\mathsf{T}\bar{\mathbf{\Omega}}\bar{\mathbf{\Omega}}^\mathsf{T}) = 2(K - \|\mathbf{V}^\mathsf{T}\bar{\mathbf{\Omega}}\|_F^2)$$
$$=: 2(K - \cos^2(\mathbf{\Theta})) = 2\sin^2(\mathbf{\Theta}) \leq 2\|\mathbf{Z}\|_F^2/\delta^2,$$

where the last inequality follows directly by the Davis-Kahan $\sin(\mathbf{\Theta})$ Theorem (Davis & Kahan, 1970; Stewart & Sun, 1990). Then

$$\|\mathbf{P}^\star - \hat{\mathbf{P}}\|_\infty \leq \|\mathbf{P}^\star - \hat{\mathbf{P}}\|_F \leq \frac{\sqrt{2}\|\mathbf{Z}\|_F}{\delta} \leq \frac{\sqrt{2K}\|\mathbf{Z}\|}{\delta}.$$

Substituting $\delta$ from (4), and defining $N := \max_k |\Omega_k^\star|$ as the size of the largest cluster, we see that

$$\|\mathbf{P}^\star - \hat{\mathbf{P}}\|_\infty \leq \frac{\sqrt{2K}\|\mathbf{Z}\|}{\delta} < \frac{\sqrt{2K}\|\mathbf{Z}\|}{\sqrt{2^3K}\|\mathbf{Z}\|\max_k |\Omega_k^\star|} = \frac{1}{2N}.$$

This implies that the difference of any two entries in $\mathbf{P}^\star$ and $\hat{\mathbf{P}}$ is bounded as

$$-\frac{1}{2N} < \mathbf{P}_{ij}^\star - \hat{\mathbf{P}}_{ij} < \frac{1}{2N} \quad\quad \forall\, i,j. \tag{7}$$

On the other hand, from (6) and the definitions of $\mathbf{\Omega}^\star$ and $\bar{\mathbf{\Omega}}$, we can see that the entries of $\mathbf{P}^\star$ are

$$\mathbf{P}_{ij}^\star := \begin{cases} \frac{1}{|\Omega_k^\star|} & \text{if } i,j \in \Omega_k^\star \\ 0 & \text{otherwise.} \end{cases} \tag{8}$$

Plugging (8) in the second inequality of (7), we see that if $i,j \in \Omega_k^\star$,

$$\hat{\mathbf{P}}_{ij}^\star > \frac{1}{|\Omega_k^\star|} - \frac{1}{2N} \geq \frac{1}{N} - \frac{1}{2N} = \frac{1}{2N}.$$

Similarly, plugging (8) in the first inequality of (7), we see that if $i$ and $j$ are not in the same $\Omega_k^\star$,

$$\hat{\mathbf{P}}_{ij}^\star < 0 + \frac{1}{2N} = \frac{1}{2N}.$$

Taking $\lambda = \frac{1}{2N}$, we see that after thresholding,

$$[\hat{\mathbf{P}}_\lambda]_{ij} = \begin{cases} \hat{\mathbf{P}}_{ij} > \frac{1}{2N} & \text{if } i,j \in \Omega_k \\ 0 & \text{otherwise.} \end{cases}$$

We thus see that the supports of $\mathbf{P}^\star$ and $\hat{\mathbf{P}}_\lambda$ are identical, which concludes the proof.

## 5 PRACTICAL RELAXATIONS

Note that achieving perfect recovery of the support of $\mathbf{P}^\star$ serves as a sufficient condition for clustering that proves to be convenient for analytical purposes. Nonetheless, it is important to recognize that this condition is not necessary and can be overly stringent in practical scenarios. Our experiments in Section 6 corroborate this fact, as they demonstrate that perfect clustering can still be achieved even when the noise level is above the requirements of Theorem 1, and even when the support of $\mathbf{P}^\star$ cannot be exactly recovered through $\hat{\mathbf{P}}_\lambda$. This outcome is both fortunate and unsurprising because, even if the support of $\hat{\mathbf{P}}_\lambda$ differs from that of $\mathbf{P}^\star$ across all values of $\lambda$ within the range $[0, 1]$, as long as the dissimilarity between $\mathbf{P}^\star$ and $\hat{\mathbf{P}}$ is not excessively large, $\hat{\mathbf{P}}$ can still contain sufficient information to unveil clustering patterns through a more lenient approach. This section outlines several candidate approaches, each with advantages and disadvantages, which we also discuss.

### $\hat{\mathbf{P}}$ AS AN ADJACENCY MATRIX

In principle, $\mathbf{P}^\star$ can be thought of as an adjacency matrix detailing the samples that belong in the same cluster. Under this lens, $\hat{\mathbf{P}}$ can be interpreted as the estimator of such adjacency matrix. Hence, we can use any machinery from the literature that estimates clusters from an adjacency matrix. One such example is spectral clustering (Ng et al., 2001), which uses a version of Lloyd's algorithm on the spectrum of the Laplacian obtained from a similarity matrix. This suggests using $\hat{\mathbf{P}}$ as a similarity matrix, computing its Laplacian, and running Lloyd's algorithm to obtain a clustering. As we will see in our experiments, this simple strategy can be very effective in practice, succeeding in noise regimes where both, the exact recovery of the support of $\hat{\mathbf{P}}$ and Lloyd's algorithm fail. Notice, however, that this induces an alternating method (Lloyd's algorithm) that is sensitive to initialization and in general can only guarantee local convergence, hence our theoretical guarantees do no apply directly to this method.

### HIERARCHICAL CLUSTERING $\hat{\mathbf{P}}$

As discussed in the previous section, some straightforward relaxations of our closed-form solution that are very effective in practice may not enjoy the same theoretical guarantees that our closed-form solution does. Fortunately, our solution is so flexible that it allows for other alternatives to which our guarantees can be extended. Here we present one example. Using the driving principle that $\mathbf{P}^\star_{ij}$ is equal to zero if samples $i$ and $j$ belong to different clusters, and nonzero otherwise, one can use $\hat{\mathbf{P}}$ as a proxy for $\mathbf{P}^\star$, and construct, bottom to top, a hierarchical tree that reveals the clustering. More precisely, since $\hat{\mathbf{P}}$ is symmetric, we would first find the entry $\hat{\mathbf{P}}_{ij}$ with the largest absolute value in the upper-diagonal (or lower-diagonal), and cluster samples $(i, j)$ together. We will iteratively repeat this process for the remaining entries in the upper-diagonal of $\hat{\mathbf{P}}$, until all the samples are clustered together. The order in which the samples are clustered will produce a tree, which at a given level will have exactly K clusters, which will be our output. The advantage of this relaxation is that it is very amicable to analysis, and one can obtain from it softer versions of Theorem 1 by exploiting existing results from the literature, such as (Dasgupta & Long, 2005). Moreover, modern techniques like (Agarwal et al., 2022) may also be adapted to obtain performance improvements in terms of space, time, and communication, which can prove particularly crucial in large-scale data analysis. Some of our future work will explore these avenues.

### ROBUST ESTIMATION OF $\mathbf{P}^\star$

Notice that one of the critical steps towards our closed-form solution is the estimation of $\mathbf{P}^\star$ through a singular value decomposition. Unfortunately, it is well-documented that this procedure is quite sensitive to outliers (Wright et al., 2009; Xu et al., 2010; Candès et al., 2011). Fortunately, there is a plethora of algorithms that may be used to overcome this difficulty (Tian & Zhang, 2022). Many of these methods have been thoroughly studied, and often enjoy theoretical guarantees and performance bounds, which can be seamlessly coupled with the Davis-Kahan $\sin(\mathbf{\Theta})$ Theorem (Davis & Kahan, 1970; Stewart & Sun, 1990) to obtain robust versions of Theorem 1.

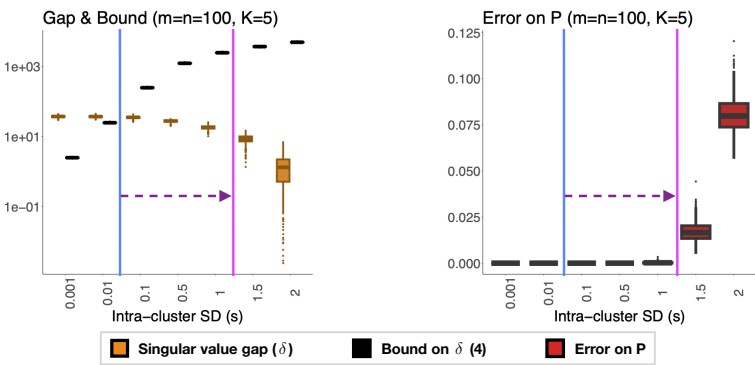

**Figure 1: Left:** Singular value gap $\delta$, which can be interpreted as the signal to noise ratio (SNR), and its lower bound (4) in Theorem 1. **Right:** Error between the supports of $\mathbf{P}^\star$ and $\hat{\mathbf{P}}_\lambda$; zero error implies that our closed-form solution recovers the true clustering perfectly. The blue line marks the value where $\delta$ equals the bound in (4). In words, Theorem 1 claims that our closed-form solution will reveal the true clustering exactly if the SNR $\delta$ is above our lower bound (left of the blue line). The figure in the right verifies this claim. The magenta line marks the largest value of $s$ (which can be interpreted as the noise level) for which the support of $\hat{\mathbf{P}}_\lambda$ is identical to that of $\mathbf{P}^\star$ (and hence the clustering can be recovered exactly). These experiments show that our closed-form solution may reveal the exact clustering even in cases where the noise is significantly higher than required by Theorem 1. The gap marked with the purple dotted arrow indicates the regime where our theory does not guarantee that our closed form solution will recover the exact clustering, yet in practice we see that it does.

## 6 EXPERIMENTS

To verify our results and test their flexibility, we ran a series of experiments on synthetic and real data. In these experiments we generated data according to (1), where the entries of each $\boldsymbol{\mu}_k^\star$ are drawn i.i.d according to a standard normal distribution, the n samples are evenly distributed among the K clusters $\{\Omega_k^\star\}$ (up to rounding error), and the entries of each $\mathbf{z}_i$ are i.i.d normal with zero mean and variance $s^2$.

### VERIFYING THEOREM 1

In our first experiment we test Theorem 1. To this end, we measured (a) the singular value gap $\delta$, (b) the bound in the right hand side of (4), and (c) the error on the estimated support of $\mathbf{P}^\star$, i.e., the average number of distinct nonzero entries in $\mathbf{P}^\star$ and $\hat{\mathbf{P}}_\lambda$, as a function of $s^2$, which represents the intra-cluster variance, related to $\|\mathbf{Z}\|$. We use a fixed $\lambda = s$. Notice that this metric is very strict, as clustering from $\hat{\mathbf{P}}$ is still possible even if the support of $\hat{\mathbf{P}}_\lambda$ is not identical to that of $\mathbf{P}^\star$ (details below). Moreover, $\lambda$ can be chosen more precisely with a sliding threshold, as discussed in Section 3. However, we wanted to test our result in its simplest form, which as we can see in Figure 1, is enough. This figure shows the results of 1000 trials for each value of $s$. Theorem 1 shows that $\mathbf{P}^\star$ can be perfectly recovered if $\delta$ satisfies the condition in (4). Figure 1 verifies this result, and suggests that the bound in (4) can be tightened, at least stochastically, if not deterministically, as $\mathbf{P}^\star$ can still be perfectly recovered when $\delta$ is outside the required range (e.g., $s \in [0.1, 1]$).

### EVALUATING PRACTICAL RELAXATIONS

In our second experiment we compare a relaxed version of our solution for the cases where the support $\mathbf{P}^\star$ cannot be perfectly estimated directly from $\mathbf{P}_\lambda$, yet $\hat{\mathbf{P}}$ still encodes enough information to reveal the clustering with any method that minimizes distortion. In these experiments we use the simplest version of Lloyd's algorithm (Lloyd, 1982) on the columns of $\hat{\mathbf{P}}$. For baseline comparison we use three canonical and prevalent methods: Lloyd's algorithm (Lloyd, 1982), K-means++ (Arthur & Vassilvitskii, 2007), and spectral clustering using the inner product matrix $\mathbf{X}^\top \mathbf{X}$ (Ng et al., 2001). As performance metric we use the number of missclassified samples after using the Hungarian algorithm (Kuhn, 1955) to find the best labels permutation. We compare these methods as a function of several parameters of the problem, namely, the intra-cluster variance $s^2$, the number of clusters K, the number of samples n, and the ambient dimension m. The results of 1000 trials for each parameter value are in Figure 2. It is clear that our approach dramatically outperforms Lloyds algorithm and K-means++. In some cases our approach outperforms spectral clustering, and in others it is marginally

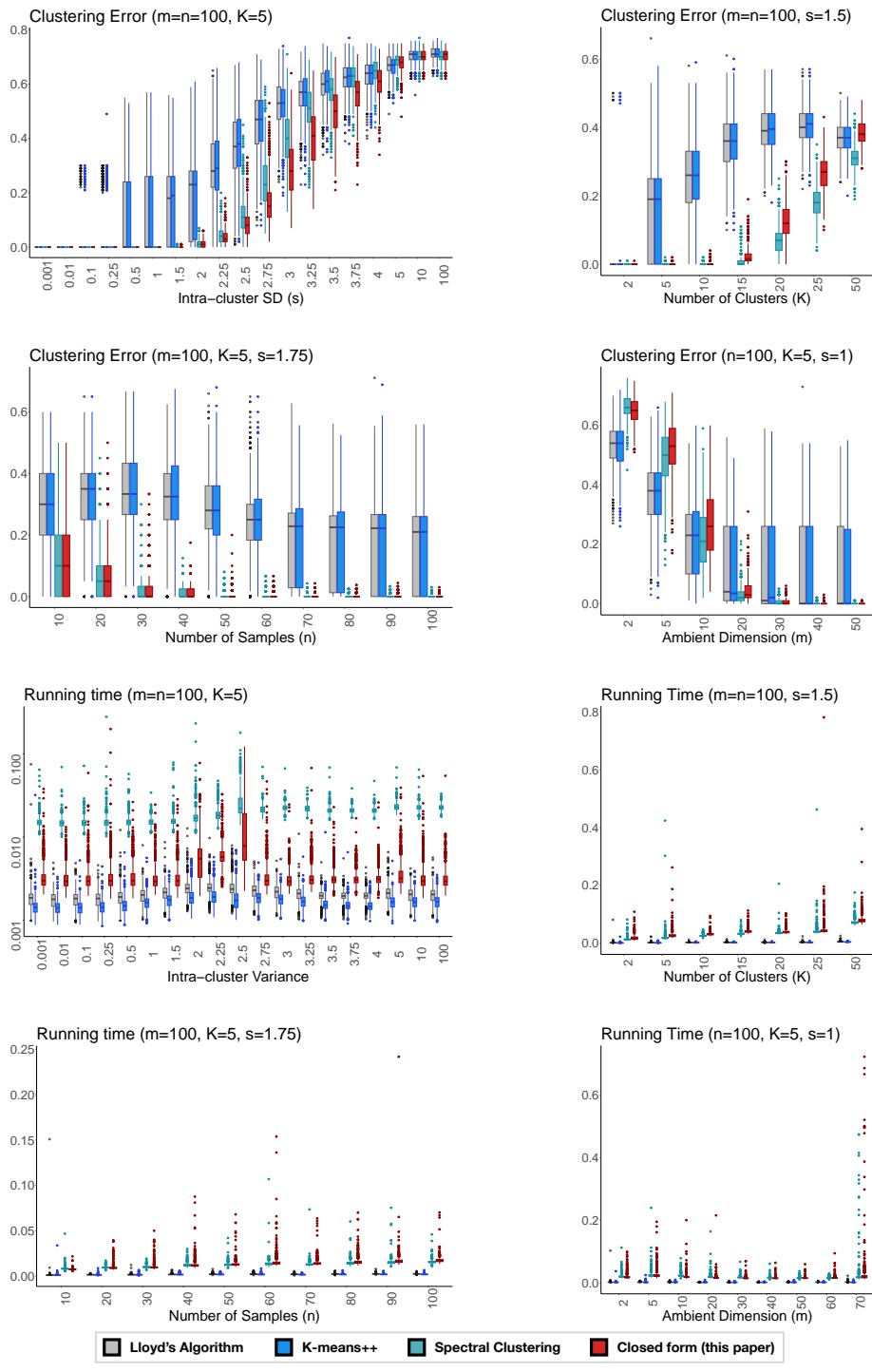

**Figure 2:** Clustering error and running time as a function of different parameters, including the intra-cluster standard deviation (s), the number of clusters (K), the number of samples (n), and the ambient dimension (m). In terms of accuracy, our approach dramatically outperforms Lloyds algorithm and K-means++, and is virtually as good as spectral clustering, yet it voids the need to construct an graphical model-based similarity matrix. Moreover, in practical settings where the ground truth is unknown, the solution of all other methods is not guaranteed to be correct, whereas ours is. In fact, these methods finish earlier, but fail to recover the true clustering (as the accuracy results show), showing that they are only converging to a local minima. In terms of running time, our solution is comparable in average to all other methods, but it has a few outlier cases, similar to spectral clustering.

| Algorithm | Buenrostro6 | Buenrostro7 | Larry | Nestorowa | Stassen |
|:---:|:---:|:---:|:---:|:---:|:---:|
| BDR | 0.4435 | 0.4031 | 0.6043 | 0.4042 | 0.696 |
| LSR | 0.3669 | 0.35 | 0.5816 | 0.4542 | **0.273** |
| AONMF | 0.3803 | 0.2271 | 0.5779 | 0.4318 | 0.485 |
| HALS | 0.4353 | 0.44 | **0.5607** | 0.3828 | 0.832 |
| ONMTF | 0.4581 | 0.4707 | 0.5682 | 0.4661 | 0.635 |
| MU | 0.4259 | 0.4433 | 0.5844 | **0.3417** | 0.615 |
| **CF (this paper)** | **0.2425** | **0.1974** | 0.5751 | 0.4505 | 0.6580 |

**Table 1:** Clustering error in several real datasets related to single-cell sequencing. The best result is highlighted in bold. Our approach significantly outperforms the state-of-the-art in several datasets. In other cases, alternative methods have comparable accuracy, yet their results cannot be certified as correct, as they lack guarantees. Only in one dataset (Stassen) our method is significantly outperformed.

worse. Our future work will investigate the exact connection between our solution and spectral clustering.

## REAL DATA EXPERIMENTS

We complement our analysis and simulations by testing our approach on five real datasets related to single-cell sequencing. We specifically selected these datasets due to the usual absence of a known ground truth in single-cell sequencing. When clustering algorithms lack solid assurances, the outcomes become questionable. Consequently, scientists frequently resort to alternative heuristic accuracy metrics like the silhouette index or the Davies-Bouldin index Ashari et al. (2023), which can yield contradictory outcomes and potentially lead to erroneous interpretations. Our research addresses this concern by offering a solution with guarantees.

All these datasets were obtained from the Gene Expression Omnibus (Edgar et al., 2002). Specifically, we used the Buenrostro 2018 data with 6 and 7 classes, the Larry dataset with 5 classes, the Nestorowa dataset with 3 classes, and the Stassen data with 10 classes. We chose these specific datasets because unlike most others in the field, their ground truth has been corroborated, so we can verify the true accuracy of each algorithm. As the name suggests, these data contain gene activation levels of a collection of cells of different types. The goal is to cluster the cells.

For baseline comparison we used comprehensive mix of classical and state-of-the-art algorithms, namely: block-diagonal representation (BDR) (Lu et al., 2018), least-squares representation (LSR) (Lu et al., 2012)alternating (AONMF) (Pompili et al., 2014), hierarchical alternating least-squares (HALS) (Shiga et al., 2016), orthogonal nonnegative matrix T-factorizations (ONMTF) (Ding et al., 2006), and multiplicative updates (MU) (Lee & Seung, 2000)

The results are summarized in Table 1, where we can see that our approach significantly outperforms the state-of-the-art in several datasets. In other cases, alternative methods show marginal improvements. However, in practice one would not be able to certify the results of these algorithms as correct, as they lack guarantees. Only in one dataset (Stassen) our method is significantly outperformed.

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
