# OpenReview forum: "Deterministic Error Bounds for Euclidean Clustering"
_ICLR.cc/2024/Conference — Submitted to ICLR 2024_

### Official Review · Reviewer_aXjr · 2023-10-31

**Soundness:** 2 fair
**Presentation:** 2 fair
**Contribution:** 2 fair
**Rating:** 5
**Confidence:** 2

**Summary:**

The Euclidean Clustering problem is a widely studied clustering problem. Based on the observation that the features
of a given clustering dataset could lie near a subspace whose projection operator encodes the clustering, in this paper, the authors give a deterministic error bound with a singular value decomposition. If the optimal clusters are "well separated", the proposed method could  deterministically guarantee the correctness of the solution. Experiments show that the proposed method achieve better performance on both synthetic and real-world datasets.

**Strengths:**

1. The proposed method is simple but effective

2. Under certain data distribution assumptions, i.e., the optimal clusters are "well separated", the proposed method gives a deterministic guarantee on the correctness of the solution

**Weaknesses:**

1. The notion of "well separated" clusters has also been widely used in Lloyd's type methods and approximation algorithms design for $k$-means clustering. Such as in [1]-[2], an $\alpha$-perturbation resilient assumption were introduced. They assume that for each optimal cluster, the data points within it are closer to each other than to the data points in other optimal clusters, which is very similar to the "not too scattered" distribution proposed in this paper. However, this paper did not give detailed analysis of time complexity and memory usage of the proposed method. It is unclear to me that whether the proposed method can be generalized to handle large-scale datasets since there already exists massively parallel $k$-means clustering method [3] that could be used to handle large-scale datasets with an optimal guarantee on clustering cost.

2. The experimental parts do not give detailed descriptions about the data sizes of the real-world datasets used in this paper.

[1] Cohen-Addad V, Schwiegelshohn C. On the local structure of stable clustering instances[C]//2017 IEEE 58th Annual Symposium on Foundations of Computer Science (FOCS). IEEE, 2017: 49-60.

[2] Angelidakis H, Makarychev K, Makarychev Y. Algorithms for stable and perturbation-resilient problems[C]//Proceedings of the 49th Annual ACM SIGACT Symposium on Theory of Computing. 2017: 438-451.

[3] Cohen-Addad V, Mirrokni V, Zhong P. Massively Parallel $k$-Means Clustering for Perturbation Resilient Instances[C]//International Conference on Machine Learning. PMLR, 2022: 4180-4201.

**Questions:**

1. Could the authors provide detailed analysis of the time and space complexities for the proposed method.

2. Could the authors compare the data distribution assumptions made in this paper for finding the optimal guarantee of the solution, with the perturbation-resilient assumptions used in approximation design for $k$-clustering problems.

3. Could the authors provide detailed data description about the sizes of the real-world datasets used in the experiments. I also wonder the scalability of  the proposed method for handling large-scale datasets, such as the million-scale datasets used in [1] and other large-scale datasets.

[1] Ren J, Hua K, Cao Y. Global Optimal K-Medoids Clustering of One Million Samples[J]. Advances in Neural Information Processing Systems, 2022, 35: 982-994.

**Details Of Ethics Concerns:**

Since this is a theoretical results, I don't this there is any ethics concerns in this paper.

---

### Official Review · Reviewer_NErf · 2023-11-01

**Soundness:** 3 good
**Presentation:** 2 fair
**Contribution:** 1 poor
**Rating:** 3
**Confidence:** 4

**Summary:**

We are given a set of points that follow some assumptions on how they are generated. In particular, there are k centers mu_1,..., mu_k, and each point is represented by adding noise to one of those k center (in that case, we say that the point belongs to the cluster of that center) The paper discusses the problem of finding the centers and what cluster each point belongs to. They show that there exists an algorithm that solves this problem given some identifiability solution (main theorem), i.e. given that the centers are enough separated, there is not too much variance in the points within a cluster, and we can observe enough points from each cluster. They discuss that this solution is also related to solve clustering problems. The algorithm is simple and based on a singular value decomposition of the data. They run extensive experiments on synthetic data for both verifying their main theorem and to show that their algorithm provides still a reasonable solution when those assumptions do not hold. They also run an analysis on five real datasets related to single-cell sequencing, showing the advantage of their methods.

**Strengths:**

The technical part of the paper is well-written and easy to follow.
I think the applications of their results to real datasets on single-cell sequencing provides significant value.

**Weaknesses:**

I think the paper fails to provide a discussion and comparison with previous work on related problems, and the relation with the clustering problem is not completely clear.

-k-means with separated data-. It seems to be known that if the data is separated, then it is possible to obtain a near-optimal clustering even using EM methods as variants of Lloyd's algorithm, see for example the papers [Ostrovsky, Rafail, et al. "The effectiveness of Lloyd-type methods for the k-means problem." Journal of the ACM (JACM) 59.6 (2013): 1-22].
Or [Jaiswal, Ragesh, and Nitin Garg. "Analysis of k-means++ for separable data." International Workshop on Approximation Algorithms for Combinatorial Optimization. Berlin, Heidelberg: Springer Berlin Heidelberg, 2012.] for k-means++

-Learning a mixture of distributions-. The paper is very similar to existing results where it is assumed that the data is generated from a mixture of distributions, and the goal is to identify the parameters of these distributions (and the distributions of the mixture that each point in their data belongs to). This problem has been extensively studied theoretically in other papers that the authors fail to cite and compare to (e.g., [1,2], note that some of this work also involves spectral methods as a singular value decomposition and a projection step). In particular, it is known that under identifiability those assumptions, it is possible to recover the parameters of these distributions (and what distribution each point belongs to).

-Relation to Clustering-. I think that the relation and the comparison with the problem of clustering is unclear.
First of all, the authors discuss the problem of k-means (or in general, center-based clustering), which is an *unsupervised* problem.
The goal of k-means clustering is to find k points, also called centers, such that the sum of the squared distance of each point in the dataset to the closest center is minimized. This is a well-defined optimization problem that has been extensively studied in the literature. This is different than the problem (1) studied by the authors.  In particular, there could be multiple optimal solutions to the k-median problem (2). It is not clear what is  "the true clustering of the samples": this creates confusion when comparing (2) and (1).

Additionally on clustering:

- In k-means clustering, the literature wants a multiplicative approximation guarantee as the optimization problem is NP-hard. "In contrast to traditional local minima guarantees ... "The k-means++ is an initialization method that provides a multiplicative approximation guarantee and not a simple heuristic.

- "This paper gives a closed-form solution to Euclidean clustering, also known as k-means clustering". This opening sentence is too strong and false: k-means clustering is a NP-hard problem so it is hard to solve it in general. They can provide a closed-form solution only under some separability assumptions, but still, their closed-form solution requires knowing what the optimal clustering is to compute the right threshold (||Z||).

- (Minor) I have never seen before the equivalence "k-means = Euclidean clustering" in the abstract.

- "In contrast to traditional local minima guarantees ", there exists a lot of literature to solve the k-mean problems that provide multiplicative error with respect to the answer (e.g.,  k-means++, or PTAS for k-means) even without any prior assumptions on the data.

- "[No practical algorithm provides correctness guarantees...]". The correctness guarantee is misleading as in center-based clustering (as k-means), we are looking to minimize an unsupervised objective, not accuracy with respect to some underlying labeling. With this interpretation, K-means++ provides a guarantee on the quality of the solution for the k-means problem, and e.g., local search heuristic provides guarantees for k-median [3].


-----

Moreover, in order to obtain the closed-formula solution of the main result, it seems that the algorithm requires the knowledge of ||Z||, meaning that it needs to know the optimal clustering to set the right threshold.


[1]: Vempala, Santosh, and Grant Wang. "A spectral algorithm for learning mixture models." Journal of Computer and System Sciences 68.4 (2004): 841-860.
[2]: Dasgupta, Sanjoy. "Learning mixtures of Gaussians." 40th Annual Symposium on Foundations of Computer Science (Cat. No. 99CB37039). IEEE, 1999.
[3]: Arya, Vijay, et al. "Local search heuristic for k-median and facility location problems." Proceedings of the thirty-third annual ACM symposium on Theory of computing. 2001.

**Questions:**

(1) Could you clarify the relation of your work with respect to existing work that identifies data generated by mixture of separated distributions as e.g. [1,2] (see Weakness)?

(2) Could you clarify the relation of your work with existing work on clustering separated data (see e.g., papers mentioned in Weakness)?

(2) Can you clarify the relation of the solution of problem (1) with k-means clustering? In particular, what happens if there are multiple optimal solution for the k-means problem?

---

### Official Review · Reviewer_CV5n · 2023-11-01

**Soundness:** 2 fair
**Presentation:** 3 good
**Contribution:** 1 poor
**Rating:** 3
**Confidence:** 3

**Summary:**

This paper introduces a novel algorithm for the well-known and well-researched k-means clustering problem. The authors demonstrate that their algorithm returns the correct clustering under certain instance structure assumptions. They also provide experimental results, validating their claim and comparing them with baselines.

**Strengths:**

The paper is well-written and the experiments consider a diverse set of synthetic and real-world datasets. The theoretical results are presented in a clear and concise manner, with all details provided.

**Weaknesses:**

I am not convinced of the novelty and importance of the results in this work. The only theoretical result is under a very strong assumption on the structure of the input points. This is in contrast to many other fast and efficient algorithms that perform well (theoretical and practical) without any assumption on the input points for k-means. As a result, the results in this work are not particularly interesting in the general setting.

Furthermore, there are a number of other works that consider well-separated instances (similar to this work) and present efficient algorithms that work deterministically. This work does not provide a clear comparison with those works, which is a significant issue. Without such a comparison, it is not clear if this work has any advantage over the known algorithms in the literature.

Finally, the experimental section is outdated. K-means is a well-studied problem, and there are even papers dedicated to comparing the performance of known algorithms on different datasets. This paper does not compare its results with the best algorithms, such as greedy k-means++ or random walk based algorithms. This is another significant issue with this work.

**Questions:**

I discussed three main weaknesses of this work, please consider providing reasons if you do not agree with them.

---

### Official Review · Reviewer_CdDR · 2023-11-05

**Soundness:** 2 fair
**Presentation:** 2 fair
**Contribution:** 2 fair
**Rating:** 3
**Confidence:** 3

**Summary:**

Authors introduced a new method for solving K-means clustering. They provided theoretical analysis that showed the error of the estimation of the assignment is bounded by the combination of number of clusters, number of samples, and variance within each cluster. Theoretical results also show that, under mild assumptions, the method is guaranteed to deterministically recover true cluster centroids and assignment. Authors also showed experimental results to support their findings.

**Strengths:**

Motivation and goal are clear. Reasoning and proofs in Section 3 and 4 look solid.

**Weaknesses:**

Although authors claim a closed form solution for the K-means clustering problem, in practice, due to its relaxation, the algorithm is not closed form anymore but requires alternating algorithms that are sensitive to initialization. Thus, theoretical guarantees in Section 3 don't apply any more.

According to Figure 2, the proposed method is slow, compared to other iterative methods. Authors didn't provide the detailed algorithm, nor its complexity. Looks like linear to K and n, but much steeper than Lloyd's.

I'm also confused by Hierarchical Clustering and Robust Estimation in page 6. They're only future possibilities. They are not concrete analysis. They seem not quite related to the rest of the paper, not contributing to any claims in the paper either.

Data used in the paper are very simple, and details of the datasets and experiments are missing, e.g. dimensionality, number of samples. Some visualizations would better help readers fully appreciate the effectiveness of the method in clustering those samples.

**Questions:**

In 6 Experiments, first paragraph: "entries of each $\pmb{\mu}^{\star}_{k}$ are draw i.i.d. according to a standard normal distribution."

First, a major issue we have in the K-means problem is that the best K is unknown. However, K is known in the synthetic data. The number of classes in the cell sequencing data, later in the section, are also known. And it seems K is known throughout the paper. I wonder when K is unknown what is the performance of the method compared to the baselines. Does it change the theories in Section 3?

Second, this normal distribution assumption is too strong for me. And there's no visualization of the data. We don't know whether the data fall into the preset that "clusters are sufficiently separated, and data in each cluster not too scattered". How does Gaussian mixture models compare to the proposed method in this case? Authors also claim that "out results guarantee ... independent of its distribution (normal, poisson, etc.)" on top of page 2, so it would be great see the results with other distributions. I wonder if authors also experimented with other types of data. What are the dimensions of the sequence data? Are they approximately normally distributed around their centroids?

Third, please discuss more about Table 1. Why proposed method outperforms most baselines in Buenrostro6 and Buenrostro7 but not in other datasets? What are the contributing factors of performing the best in each dataset? I see AONMF is also consistently good across all datasets, but proposed CF is pretty moderate in Larry, Nestorowa, and Stassen.

---

### Meta-Review · Area_Chair_9jeu · 2023-12-04

**Metareview:**

The main issue with this paper is that there are numerous theoretical and empirical studies of k-means.  The results of this paper make very strong assumptions on the input points and then derive results.  It was unclear what the novelty is of this paper's over prior work and the reviewers found the comparison to prior work lacking.

**Justification For Why Not Higher Score:**

No reviewer was excited. The paper fails to provide a solid comparison to prior work.

**Justification For Why Not Lower Score:**

N/A

---

### Decision · Program_Chairs · 2024-01-16

Reject